# Knowledge Assessment of E-Bug Assisted Antimicrobial Resistance Education Module in Class VII School Students of South Indian Coastal Town of Manipal

**DOI:** 10.3390/jcm8010084

**Published:** 2019-01-12

**Authors:** Reona Fernandes, Swathi Naik, Archana-Gururaj Bhat, Rashmi Shetty, Manjunatha-H Hande, Abdul Ghafur, Mahadev Rao, Vijayanarayana Kunhikatta, John-Preshanth-Kumar Pathiraj

**Affiliations:** 1Department of Pharmacy Practice, Manipal College of Pharmaceutical Sciences, Manipal Academy of Higher Education, Manipal 576104, India; fernandesreona@gmail.com (R.F.); swathikn2496@gmail.com (S.N.); archanagururajbhat8@gmail.com (A.-G.B.); rashmishetty1199@gmail.com (R.S.); mahadev.rao@manipal.edu (M.R.); vijayanarayana.k@manipal.edu (V.K.); 2Department of Medicine, Kasturba Medical College and Hospital, Manipal Academy of Higher Education, Manipal 576104, India; manjunath.hande@manipal.edu; 3Infectious Diseases Department, Apollo Cancer Institute, 320 Anna Salai, Chennai 600035, India; drghafur@hotmail.com

**Keywords:** antimicrobial resistance, stewardship, community, school, students, e-bug, education, pharmacists, India

## Abstract

Antimicrobial resistance (AMR) is a recognized public health threat today globally. Although many active and passive stewardship strategies are advocated to counter AMR clinically, educating school going children on AMR could be a cost-effective measure to minimize AMR development in the future. We implemented NICE’s e-bug as a module to educate class VII school students on AMR determinants. A prospective quasi-experimental study on 327 students from nine different schools of class VII around Manipal town, Udupi district, Karnataka state, India were included in the study. Ten questions on AMR determinants from the e-bug program were used in written pre-test. After an education intervention, a post-test was conducted. Descriptive statistics to estimate epidemiological characteristics, Wilcoxon Signed Ranks and Kruskal–Wallis tests were applied to analyze statistical significance of pre/post-test performance scores and between schools. Students had inadequate knowledge on seven AMR determinants (antimicrobial indication, its course, hand hygiene, fermentation, spread of infection, microbial multiplication and characteristics of microbe) as analyzed from the post-test performance (*p* < 0.05). Comparison of post-test performance between schools showed significant improvement in scores (*p* < 0.05) for three questions (definition on antimicrobial, cover while cough/sneezing and microbial characteristics). Although students exhibited sub-optimal knowledge on some AMR determinants, they showed keenness to learn, which was evident by their post-test performance. Our findings and previous similar studies from Europe are suggestive of early pedagogic interventions on AMR through inclusion of such education modules in the regular school curriculum could be a potential tool for AMR prevention.

## 1. Introduction

Antimicrobial resistance (AMR) is recognized as a public health hazard worldwide [1]. Policy makers, healthcare workers and other stakeholders have advocated various methods to counter AMR [2]. Studies have tied AMR to inappropriate antimicrobial use encountered across the entire spectrum of healthcare settings [3]. Antimicrobial stewardship programs (ASP) initiated by expert infectious disease physicians and pharmacists are a recognized strategy by the World Health Organization (WHO), European Centre for Disease Control (ECDC), Centre for Disease Control (CDC) etc., to counter AMR. Although many clinical and lab based active and passive interventions have been in practice, behavioral changes through educational interventions targeted towards students of various age groups are reported to be beneficial [4]. ASP at the community level is sparsely reported, although AMR burden is commonly seen in this setting. We identified a study that reports community pharmacist-led ASP leads to a reduction in inappropriate antimicrobial prescribing [5].

### Global Initiatives against AMR and Education as Stewardship Strategy in Children from an Indian Viewpoint

ASP in India is still in its nascent stage since the Chennai declaration [6,7]. The physician–patient ratio is abysmally low, especially in rural areas [8]. This shortcoming could be addressed by allied health professionals such as pharmacists particularly in community health education [9,10]. Educational interventions on health aspects aimed at school going children have been reported to impart healthier lifestyle [11]. Promoting knowledge through hands-on educational programs in children is reported to improve awareness about infection prevention and antimicrobial use [12,13]. Many government funded programs such as the ECDC’s “European Antibiotic Awareness Day”, United Kingdom’s National Institute of Clinical Excellence (NICE) e-bug program, France’s “Microbes en question”, Canada’s “Do Bugs Need Drugs?”, and the CDC’s “Get Smart about Antibiotics” developed in Europe and North America emphasize educating school children about antimicrobials and infection [14]. It is worth mentioning that, although such programs exist, we have yet to come across the inclusion of such tools and models as a part of the mainstream academic curricula for school going children. Particularly school children are considered to be vulnerable for high rate of infections and frequent antimicrobial use [15]. Prescribing antimicrobials in children is also associated with their parents’ expectations and beliefs thereby influencing prescribers [15]. A previously reported Indian study qualitatively assesses AMR knowledge among school students and teachers without any interventions or post-assessment [16]. Since there was no report on any experimental study in India, we conducted a pre-post intervention study targeted at junior school children.

The objective of this education session was to serve as a novel strategy towards inculcating knowledge and encourage active learning as a stewardship strategy for AMR prevention among school children.

## 2. Materials and Methods

A prospective quasi-experimental pre-post study was conducted on class VII school children from select schools around the southwest of Karnataka state (Manipal Town, Udupi District, Karnataka State), India during August 2017 to April 2018. The study was approved by the Institutional Ethics Committee (IEC), Kasturba Hospital, Manipal [ref no. KH IEC: 676/2017].

### 2.1. Selection of Study Participants and Criteria

Eligible class VII students aged 11–13 years from public, private and charitable schools participated in the study after assenting from their parents (Figure 1). One school that acknowledged participating but was not permitted by the IEC (due to the scheduled status given by the government) was not included in the study. Parents who did not assent and those parents who assented but whose children were not present during the education session were excluded. During the assenting process, few queries based on the details provided in the “participant information sheet” were acknowledged telephonically. The queries were: one parent (a) enquired about the study, few parents enquired seeking specific information, such as (b) whether the study involved giving some injections to their children, (c) whether any medications would be given to their children, (d) whether they have to miss their classes to participate, and (e) if they have to travel outside the school for participating in the study.

### 2.2. Educational Intervention Module

We visited the schools twice and the entire study was executed during routine school hours. Study participants filled the study quiz in English and Kannada. The first visit was submission of the informed consent document and subject information for assenting from the student’s parents for participation in the study. During the second visit, the education intervention was performed. The entire education session was designed for 40 min duration in the class room provided with a pre-test (10 min) followed by an educational intervention (for 20 min) and a post-test (10 min). The pre-test was based on ten questions from e-bug database for junior grade children with a written quiz on factors contributing to AMR. Medium of instruction was both English and Kannada. The questions covered concepts on microbes, antimicrobials, AMR, hygiene and fermentation process (Figure 2). Pre-test was followed by an education intervention and an interactive session. Education also involved the use of education aids such as hand drawn images and models representing microbes and other educational content. The study intervention concluded with a post-test quiz with the same questionnaire that was used during pre-test.

### 2.3. Statistical Analysis

Marked pre- and post-test responses were manually evaluated. Each correct response was given a point. The percentage scores were calculated at the school level and for the whole cohort. Individual student computed scores were entered first in Microsoft Excel 2010 sheets, and then exported to SPSS Version 20 (IBM Corp., Chicago, IL, USA) for further statistical analyses. Computation of Overall percent (correct) was computed as: (Sum of students with correct responses from School 1 to 9/Total number of participating students) × 100. For each AMR determinant (question), Wilcoxon signed Rank and Kruskal–Wallis tests were applied to analyze pre- and post-test responses of students and overall participating schools, respectively. *p*-value of *p* < 0.05 was considered statistically significant.

## 3. Results

Our study covered nine schools, in which a total of 327 students participated and completed the education session. Overall, 160 (49%) students were boys and 167 (51%) were girls (Table 1). They all belonged to VII grade. Five schools had English as a language medium of education and four schools had Kannada. Students were educated and tested in their respective medium of instruction. Of the nine schools, four were state government run, four were private institutions and one charitable. Syllabi-wise, two were affiliated to Central Board for Secondary Education (CBSE), Government of India and seven belonged to state board, Karnataka state. We used ten questions published in the e-bug program database aimed for AMR education in school children. Students’ knowledge for questions (Number 1, 2, 5, 7, 8, 9 and 10 were based on definition on antimicrobial, general characteristics of microbe, uses of microbes, spread of infection and personal hygiene, respectively) showed statistical significance (*p* < 0.05) with our education intervention (Table 2). Comparison of the post-test students’ performance from the nine participating schools are presented in Table 3. Responses for Questions 1, 3 and 10 showed significant inter school variability (*p* < 0.05).

## 4. Discussion

Children are believed to be more creative, which helps them enhance active learning. E-bug, a pan-European antimicrobial and hygiene teaching resource, is reported to serve as an educational tool aimed to create awareness in school children and teachers on antimicrobials, hygiene and transmission of infections [11]. We tested the e-bug module for junior school going children in our study school children (Table 2) and compared the post-test inter school performance scores (Table 3).

Our study is a unique one from our country, since we assessed the baseline knowledge junior school students and teachers and post-test improvement, unlike a previous Indian study [16] that reports only the baseline awareness. We identified one with similar study design previously reported from Portugal on ninth grade students from two different schools, one in an urban setting and the other in a rural setting, using a predefined set of questions on antimicrobial for bacterial infections. They however did not cover questions from areas of immunity, hand hygiene and microbes [17]. We followed 10 questions from the e-bug program as a module to cover topics on antimicrobial use and various AMR determinants. For a question on completing the course of antimicrobial therapy, there was a marked improvement in scores between pre- and post-tests (*p* < 0.05). Lack of awareness on antimicrobial regimen course completion in 9th and 12th grade students is also reported by another study from Portugal [12]. The question on hand hygiene showed significant improvement during post-test (*p* < 0.001), highlighting the role training can have on the key knowledge and possibly behavior of school children. The question attempted to improve awareness of personal hygiene to keep infections away. Inadequate knowledge on handwashing is reported in a cross-sectional study among 8–9-year-old children belonging to an urban school in Mumbai, India [18]. The term bacteria or microbe is invariably understood as something harmful or dangerous. We explored the knowledge about the usefulness of non-pathogenic microbes in making bread and yogurt through fermentation. Students were “surprised” to know the uses of microbes indicative in their positive responses (*p* < 0.001). A small, interview-based French qualitative study corroborates the knowledge of school children on uses of microbes. The findings reveal that students cannot state the applications of microbes in biotechnology field [19].

Our findings show 48.6% of students were not aware that microbes can be picked up from door handles when a question on source of infection was posed. Post education, their knowledge showed significant improvement (*p* < 0.001). Microbial contamination of door handles and knobs are well documented vehicles for cross-infections and recontamination of washed hands resulting in spread of infection between persons. A study from Nigeria reports door handles and knobs of toilets and bathrooms in churches, markets/parks, banks, restaurants and government establishments tested positive for bacterial contaminants [20].

We analyzed children’s knowledge on microbial multiplication showing significant improvement following the education session (*p* < 0.001). This was in line with an earlier study reporting on children 7–14 years of age [21]. They were educated by sketching an image of bacterial multiplication on the black board, i.e. a picture of how microbes can multiply rapidly. Children of different age group possess different ideas about microbes. Additionally, the education on microbes included microorganism models depicting general characteristics of microbes, e.g., size, various common infectious diseases and health, which were discussed using appropriate models. Children considered reproduction of microbes to be an aggressive process, which, according to them, was dangerous. The post-test knowledge score on microbes being living organisms that cannot be seen through a naked eye was significantly better than pre-test (*p* < 0.001) suggesting the need for this basic education among children. This corroborates to an earlier interview-based finding by M. Gail Jones et al. on 5th-, 8th-, and 11th-grade students, teachers, and medical professionals using a pre-designed drawings of morphological characteristics and various types of microorganisms. They concluded students, teachers and medical professionals had different understandings about microbial characteristics. The authors, however, emphasized the need for education at different levels of academic and professional qualifications [22].

There are few limitations to our study. The study was conducted using convenient sampling approach by involving schools from a geographical location adjoining Manipal Town, Udupi District, India. The findings are to be interpreted for this region, which is acclaimed for better education standards in India. The education intervention was conducted wholly by trained final year undergraduate pharmacy (four-year course) students, hence the findings could vary if performed by an infectious disease expert. The Kannada (questions) version translation was done by the authors and was not validated by a language expert. Lastly, attributes surrounding inter-school post-test performance could not be explored as reported by the Portuguese findings [12], due to heterogeneity of students, which we also considered to be beyond the scope of our stated objectives.

Our findings reiterate earlier findings that educating children during middle to high school on awareness on infection, microbes and inculcate scientific interest towards shaping behavior on appropriate antimicrobial use is needed [23,24,25]. Understanding implications of non-pathogenic/pathogenic bacterial effects and personal (hand) hygiene education needs to be addressed among school going students. Follow-up knowledge tests of the study cohort will provide an insight about mid- to long-term recall of the education session which could be assessed after a wash-out period of one year. Similar implementation of educational module in regular school curriculum promises to be a cost-effective strategy to counter AMR with continuous regulatory support in the future.

## 5. Conclusions

Knowledge on core components that contribute to AMR was found to be insufficient in middle-school children in India. The educational intervention improved the awareness significantly. Students showed keen interest in learning concepts of antimicrobial use and microbes and their post-test performance suggests good understanding, which needs to be utilized effectively as novel community ASP strategy. The present study provided insight into the knowledge of junior school children on antimicrobials and AMR and the impact of education intervention on their knowledge.

## Figures and Tables

**Figure 1 jcm-08-00084-f001:**
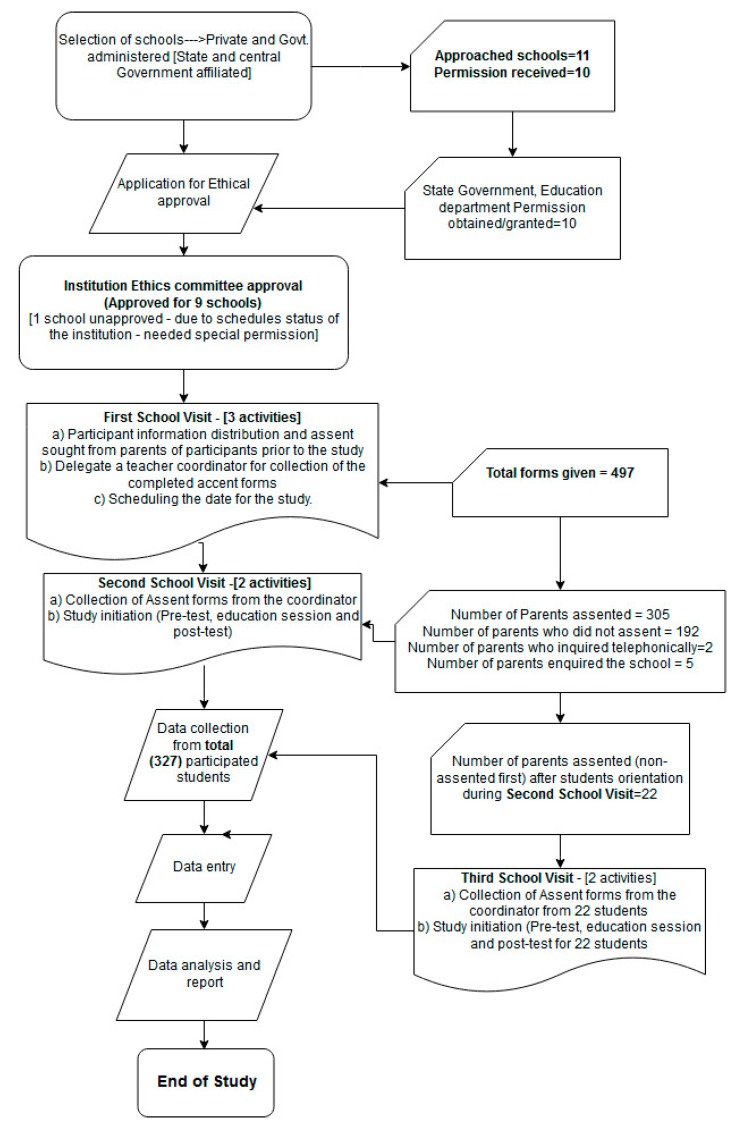
Overview of study methodology.

**Figure 2 jcm-08-00084-f002:**
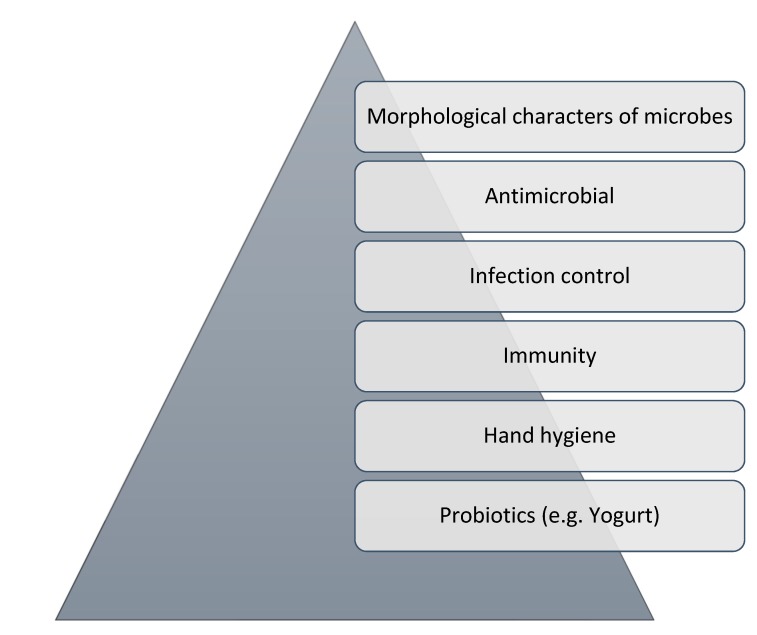
Topics covered during educational intervention.

**Table 1 jcm-08-00084-t001:** Descriptive of students and school demographic data.

Description		*N*	%
Total Schools (9)	School 1 students	87	26.6
	School 2 students	29	8.9
	School 3 students	14	4.3
	School 4 students	11	3.4
	School 5 students	40	12.2
	School 6 students	47	14.4
	School 7 students	14	4.3
	School 8 students	27	8.3
	School 9 students	58	17.7
Gender (overall)	Male	160	48.9
	Female	167	51.1
School Type	Private	221	67.6
	Government	81	24.8
	Charitable	25	7.6
School Syllabus	State	251	76.8
	CBSE	76	23.2

*N* = frequency of school students participated, % = percentage of school students participated, CBSE = Central Board of Secondary Education.

**Table 2 jcm-08-00084-t002:** Comparison of pre and post percentage scores of participating students.

Questions		Correct (%)	Incorrect (%)	*p*-Value ^@^
Q1: Antimicrobials are used to kill bacteria	Pre	85.0	15.0	**<0.001 ***
	Post	93.9	6.1	
Q2: You should always finish your course of antimicrobials	Pre	77.1	22.9	**<0.001 ***
	Post	93.0	7.0	
Q3: We should always cover our coughs and sneezes	Pre	92.0	8.0	**0.869**
	Post	92.4	7.6	
Q4: All microbes on our hands are good for us	Pre	92.0	8.0	**0.086**
	Post	95.1	4.9	
Q5: We should only wash our hands once a day	Pre	92.4	7.6	**0.011 ***
	Post	96.0	4.0	
Q6: Some microbes can make us ill	Pre	89.3	10.7	**0.238**
	Post	91.7	8.3	
Q7: We use some microbes to make bread and yogurt	Pre	49.8	50.2	**<0.001 ***
	Post	84.1	15.9	
Q8: You can pick-up microbes from door handles	Pre	51.4	48.6	**<0.001 ***
	Post	75.2	24.8	
Q9: Microbes can multiply very fast	Pre	84.1	15.9	**<0.001 ***
	Post	95.1	4.9	
Q10: If you cannot see a microbe, it is not there.	Pre	70.6	29.4	**<0.001 ***
	Post	85.0	15.0	

^@^ Wilcoxon Signed Ranks Test was applied to compute pre- and post-test percentage scores, *p* < 0.05 was considered statistical significant, those in boldface indicate statistical significant. * Questions that were found to be statistically significant.

**Table 3 jcm-08-00084-t003:** Frequency of correct answers school-wise and overall post-test percentage response of participating schools.

Questions	School 1 (*n*)	School 2 (*n*)	School 3 (*n*)	School 4 (*n*)	School 5 (*n*)	School 6 (*n*)	School 7 (*n*)	School 8 (*n*)	School 9 (*n*)	Overall % (Correct) ^$^	*p*-Value
Q1	86	29	14	10	28	45	14	27	54	93.9	<0.001 ^#^
Q2	82	27	14	11	34	46	12	26	52	93.0	0.248
Q3	84	27	14	10	28	46	13	27	53	92.4	<0.001 ^#^
Q4	83	28	14	9	36	46	14	27	54	95.1	0.223
Q5	83	29	14	9	39	45	12	27	56	96.0	0.114
Q6	77	27	12	11	37	46	13	24	53	91.7	0.692
Q7	66	28	13	11	33	41	13	25	45	84.1	0.057
Q8	59	22	11	10	42	32	12	24	42	75.2	0.191
Q9	81	27	14	11	39	47	12	24	56	95.1	0.243
Q10	72	27	11	9	31	47	8	26	47	84.0	<0.001 ^#^

In the above table, *n* = frequency of participating students. ^$^ overall percent (correct) = (Sum of students with correct responses from School 1 to 9/Total number of participating students) × 100. Kruskal–Wallis Test was applied to compare inter school posttest percentage response. ^#^ Post-test responses that showed statistical significance (*p* < 0.05).

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
