# Peer review of "Knowledge Assessment of E-Bug Assisted Antimicrobial Resistance Education Module in Class VII School Students of South Indian Coastal Town of Manipal"

_jcm, 2019, doi:10.3390/jcm8010084_

Reviewer 1 Report

Resistance to antibiotics is undoubtedly a major public health problem. Awareness of the population and school-age children is very important in combating inappropriate use of antibiotics and the development of bacterial resistance.

Although the manuscript focuses on a major problem, the approach, the methodology and the analysis that is done do not seem to have enough scientific rigor to draw conclusions.

Author Response

Dear Reviewer,

We thank and appreciate for the critiquing the manuscript and helping us to improve the quality of presentation. We, the authors of this manuscript have accepted all the comments and have responded to the comments and modified the manuscript accordingly. On reviewing the manuscript we sincerely hope to have made significant changes and improved the content.

Table 2 we have done left alignment for better presentation.

We would like to bring to our reviewers notice that, Table 3 is modified in accommodate with frequency of each schools posttest responses while we have retained the same format as presented in the earlier version. Formula for computation of the overall % score is written as a table legend and also included in the methodology.

Another correction made in Table 2. Q10 post-test percentage, corrected from the earlier value (from 84 to 85) which was a typographical error.

Sincerely,

Authors

Review comments: 

Resistance to antibiotics is undoubtedly a major public health problem.

Awareness of the population and school-age children is very important in combating inappropriate use of antibiotics and the development of bacterial resistance.

Comment1

Although the manuscript focuses on a major problem, the approach, the methodology and the analysis that is done do not seem to have enough scientific rigor to draw conclusions.

General response 1:

We appreciate and thank for supporting the study in principle and the critique. We have cited the rationality and objectives of the study to improvise the introduction portion of the manuscript.  The authors believe that the study findings and its analysis justifies the now modified introduction and objective of the study.   

Specific response 1:

 The approach to use school education as a novel stewardship tool is adapted from programs reported from a spectrum of nations globally.

Specific response 2:

 As our study is a prospective pre-post education study we have followed the prescribed method as published in earlier studies (cited.. Maria-Manuel Azevedo et.al 2013) and as per IEC recommendations/suggestion provided for community school children.

Specific response 3:

We have tried to present the findings based on our objective and discuss them. By implementing the e-bug module, we intended to test baseline children’s knowledge on AMR and tested the impact of education intervention. We believe that our results is to be interpreted in line with other earlier reported similar findings and stewardship initiatives reported globally.

Reviewer 2 Report

Thank you for allowing me to review the following manuscript: 

Assessment of e-bug database assisted education of class VII school children on antimicrobial resistance  determinants: a non-randomized education study in a  cross-section of schools around Manipal town,  Udupi, India

I have following compulsory revisions for the authors:

Title is too long please improve

Abstract line 15 needs to be re-written

Most references are incomplete and need to be in journal style

Line 43 use full forms

Line 60 the word “hypothesized” is wrong, suggestion is to use “implemented”

Line 70 to 75 should be written in a better manner. There is no need of questions in the text.

Line 82 Medium of the written quiz and education was in English and Kannada. Please specify- if the education was given in English in a English medium school and in Kannada in a vernacular school OR a mix of two language was used?

Lines 137-138 needs to be rewritten

Lines 147-149 are unclear and need a reference at the end of sentence.

Lines 170-177 need reference

 An English language review will greatly benefit the manuscript's readability.

Author Response

Dear Reviewer,

We thank and appreciate for the critiquing the manuscript and helping us to improve the quality of presentation. We, the authors of this manuscript have accepted all the comments and have responded to the comments and modified the manuscript accordingly. On reviewing the manuscript we sincerely hope to have made significant changes and improved the content.

Table 2 we have done left alignment for better presentation.

We would like to bring to our reviewers notice that, Table 3 is modified in accommodate with frequency of each schools posttest responses while we have retained the same format as presented in the earlier version. Formula for computation of the overall % score is written as a table legend and also included in the methodology.

Another correction made in Table 2. Q10 post-test percentage, corrected from the earlier value (from 84 to 85) which was a typographical error.

Sincerely,

Authors

Review comments: 

I have following compulsory revisions for the authors:

Comment1

Title is too long please improve

Previous title: Assessment of e-bug database assisted education of class VII school children on antimicrobial resistance determinants: a non-randomized education study in a cross-section of schools around Manipal town, Udupi, India

Response1

We appreciate the suggestion on shortening the title.  With modifications made we have retained the geography of our study site. 

Modified Title: Knowledge assessment of e-bug assisted antimicrobial resistance education module in class VII school students of south Indian coastal town of Manipal..

Comment2

Abstract line 15 needs to be re-written

response2

Word hypothesised changed to implemented

Comment3

Most references are incomplete and need to be in journal style

response3

Styled as per journal recommendations

Comment4

Line 43 use full forms

R2response4

Done with full forms

Comment5

Line 60 the word “hypothesized” is wrong, suggestion is to use “implemented”

response5

Word hypothesised was modified to another better term ‘conducted’

R2Comment6

Line 70 to 75 should be written in a better manner. There is no need of questions in the text.

response6

Line 70 to 75 rewritten and the questions in the text converted into points

R2Comment7

Line 82 Medium of the written quiz and education was in English and Kannada. Please specify- if the education was given in English in a English medium school and in Kannada in a vernacular school OR a mix of two language was used?

response7

Line 82 language medium clarified.

Comment8

Lines 137-138 needs to be rewritten

response8

Lines 137-138 rewritten

Comment9

Lines 147-149 are unclear and need a reference at the end of sentence.

response9

Lines 147-149 made clear and reference cited

Comment10

Lines 170-177 need reference

response10

Lines 170-177 reference cited

Comment11

An English language review will greatly benefit the manuscript's readability

response11

We accept the recommendation on English language improvement. We copyedited a significant portion of the manuscript with assistance from a dept. colleague.

Reviewer 3 Report

In the manuscript „Assessment of e-bug database assisted education of class VII school children on antimicrobial resistance determinants: a non-randomized education study in a cross-section of schools around Manipal town, Udupi, India” the authors reported on a non-randomized intervention study on more than 300 students of schools class VII around Manipal town, India. The authors used questions on AMR determinants as pre-test followed by an education intervention on the same questions followed by a post-test to end the session. The results were statistically analysed. The analysis of the post-test performance showed that the students had inadequate knowledge on AMR determinants. The authors suggested that early pedagogic interventions on AMR could be potential tool for AMR prevention for future generations.

The manuscript is interesting and well written. However, this article will only reach a very limited group of interested readers. This reviewer feels that the introduction needs more information, as addressed below, to be interesting for a broad readership. However, the reviewer also misses some striking suggestions/discussion on the pro’s and con’s of such pre-education studies. 

Introduction should be extended to explain its global impact. This reviewer also misses an introduction into the suitability of such “pre-education” studies in other countries…

Line 42: please include some examples for clinical antimicrobial stewardship programs (ASP)

Line 43: WHO, ECDC, CDC: Abbreviation should be explained

Line 46: Reference needs to be re-place before dot

Line 58/60: Please delete the space between [13] and . & [14] and .

Line 68: Change Figure.1 to Figure 1

Line 82: please adapt the text style

Line 94: Change Figure.2 to Figure 2

Line 97 Change eg: to e.g. (also in Figure 2 Probiotics…)

Line 114: Change Table-1 to Table 1

Line 121: intervention(Table.2) A space is missing and delete the dot

Line 122: Change Table.3 to Table 3

Line 136: (3) [Table.2 and 3] please shift this before the dot and change Table.3 to Table 3

Please revise the manuscript carefully to match all minor mistakes in the revised submission…

Author Response

Dear Reviewer,

We thank and appreciate for the critiquing the manuscript and helping us to improve the quality of presentation. We, the authors of this manuscript have accepted all the comments and have responded to the comments and modified the manuscript accordingly. On reviewing the manuscript we sincerely hope to have made significant changes and improved the content.

Table 2 we have done left alignment for better presentation.

We would like to bring to our reviewers notice that, Table 3 is modified in accommodate with frequency of each schools posttest responses while we have retained the same format as presented in the earlier version. Formula for computation of the overall % score is written as a table legend and also included in the methodology.

Another correction made in Table 2. Q10 post-test percentage, corrected from the earlier value (from 84 to 85) which was a typographical error.

Sincerely,

Authors

Review comments: 

The manuscript is interesting and well written. However, this article will only reach a very limited group of interested readers.

This reviewer feels that the introduction needs more information, as addressed below, to be interesting for a broad readership.

Comment1

The reviewer also misses some striking suggestions/discussion on the pro’s and con’s of such pre-education studies.

response1

The authors accept the reviewers comment on pre-education. We have made efforts to present the need for educating junior school students by justifying as per the cited literature in introduction and discussion part of the manuscript.

Comment2

Introduction should be extended to explain its global impact.

response2

We again thank the reviewer’s observation on the need for the study in Indian sub-special-population which we have introduced in the global context and past report from India.

Comment3

This reviewer also misses an introduction into the suitability of such “pre-education” studies in other countries…

response3

The suitability of pre-education studies as highlighted by the reviewer, the authors’ response for comment1 and comment2 to be considered.

Comment4

Line 42: please include some examples for clinical antimicrobial stewardship programs (ASP)

response4

Line 42: We appreciate the reviewer’s observation on the need to list various ASP strategies. The authors however believe that AMR is an established reality and ASPs being an evidence based practice reported within various categories of healthcare professionals. We, hence have avoided listing various ASP strategies. However, we have cited Céline Pulcini et.al review attempting to merge ASPs with a special need for education as an useful strategy.

Comment5

Line 43: WHO, ECDC, CDC: Abbreviation should be explained

response5

Line 43 – Abbreviations expanded

Comment6

Line 46: Reference needs to be re-place before dot

response6

Line 46 - corrected

Comment7

Line 58/60: Please delete the space between [13] and . & [14] and .

response7

Line 58/60 – Space deleted

Comment8

Line 68: Change Figure.1 to Figure 1

response8

Line 68: formatted

Comment9

Line 82: please adapt the text style

response9

Line 82: Text style formatted

Comment10

Line 94: Change Figure.2 to Figure 2

R3response10

Line 94: formatted

Comment11

Line 97 Change eg: to e.g. (also in Figure 2 Probiotics…)

response11

Line 97 formatted

Comment12

Line 114: Change Table-1 to Table 1

R3 response12

Line 114: formatted

Comment13

Line 121: intervention(Table.2) A space is missing and delete the dot

response13

Line 121: formatted

Comment14

Line 122: Change Table.3 to Table 3

response14

Line 122: formatted

Comment15

Line 136: (3) [Table.2 and 3] please shift this before the dot and change

response15

Line 136: formatted

Comment16

Table.3 to Table 3

response16

Formatted

Please revise the manuscript carefully to match all minor mistakes in the revised submission…

Round  2

Reviewer 2 Report

The authors have addressed my comments. I have no further comments to offer.

Reviewer 3 Report

The quality of the revised manuscript has been improved.